# Modeling Tabular Data using Conditional GAN

**Lei Xu**
MIT LIDS
Cambridge, MA
leix@mit.edu

**Maria Skoularidou**
MRC-BSU, University of Cambridge
Cambridge, UK
ms2407@cam.ac.uk

**Alfredo Cuesta-Infante**
Universidad Rey Juan Carlos
Móstoles, Spain
alfredo.cuesta@urjc.es

**Kalyan Veeramachaneni**
MIT LIDS
Cambridge, MA
kalyanv@mit.edu

## Abstract

Modeling the probability distribution of rows in tabular data and generating realistic synthetic data is a non-trivial task. Tabular data usually contains a mix of discrete and continuous columns. Continuous columns may have multiple modes whereas discrete columns are sometimes imbalanced making the modeling difficult. Existing statistical and deep neural network models fail to properly model this type of data. We design CTGAN, which uses a conditional generator to address these challenges. To aid in a fair and thorough comparison, we design a benchmark with 7 simulated and 8 real datasets and several Bayesian network baselines. CTGAN outperforms Bayesian methods on most of the real datasets whereas other deep learning methods could not.

## 1    Introduction

Recent developments in deep generative models have led to a wealth of possibilities. Using images and text, these models can learn probability distributions and draw high-quality realistic samples. Over the past two years, the promise of such models has encouraged the development of generative adversarial networks (GANs) [10] for tabular data generation. GANs offer greater flexibility in modeling distributions than their statistical counterparts. This proliferation of new GANs necessitates an evaluation mechanism. To evaluate these GANs, we used a group of real datasets to set-up a benchmarking system and implemented three of the most recent techniques.

Table 1: The number of wins of a particular method compared with the corresponding Bayesian network against an appropriate metric on 8 real datasets.

|                     | outperform | |
| --- | --- | --- |
| Method | CLBN [7] | PrivBN [28] |
| MedGAN, 2017 [6] | 1 | 1 |
| VeeGAN, 2017 [21] | 0 | 2 |
| TableGAN, 2018 [18] | 3 | 3 |
| CTGAN | **7** | **8** |

For comparison purposes, we created two baseline methods using Bayesian networks. After testing these models using both simulated and real datasets, we found that modeling tabular data poses unique challenges for GANs, causing them to fall short of the baseline methods on a number of metrics such as *likelihood fitness* and *machine learning efficacy* of the synthetically generated data. These challenges include the need to simultaneously model discrete and continuous columns, the multi-modal non-Gaussian values within each continuous column, and the severe imbalance of categorical columns (described in Section 3).

To address these challenges, in this paper, we propose conditional tabular GAN (`CTGAN`)[1], a method which introduces several new techniques: augmenting the training procedure with *mode-specific normalization*, architectural changes, and addressing data imbalance by employing a *conditional generator* and *training-by-sampling* (described in section 4). When applied to the same datasets with the benchmarking suite, `CTGAN` performs significantly better than both the Bayesian network baselines and the other GANs tested, as shown in Table 1.

The contributions of this paper are as follows:

**(1) Conditional GANs for synthetic data generation**. We propose `CTGAN` as a synthetic tabular data generator to address several issues mentioned above. `CTGAN` outperforms all methods to date and surpasses Bayesian networks on at least 87.5% of our datasets. To further challenge `CTGAN`, we adapt a variational autoencoder (VAE) [15] for mixed-type tabular data generation. We call this `TVAE`. VAEs directly use data to build the generator; even with this advantage, we show that our proposed `CTGAN` achieves competitive performance across many datasets and outperforms `TVAE` on 3 datasets.

**(2) A benchmarking system for synthetic data generation algorithms**.[2] We designed a comprehensive benchmark framework using several tabular datasets and different evaluation metrics as well as implementations of several baselines and state-of-the-art methods. Our system is open source and can be extended with other methods and additional datasets. At the time of this writing, the benchmark has 5 deep learning methods, 2 Bayesian network methods, 15 datasets, and 2 evaluation mechanisms.

## 2 Related Work

During the past decade, synthetic data has been generated by treating each column in a table as a random variable, modeling a joint multivariate probability distribution, and then sampling from that distribution. For example, a set of discrete variables may have been modeled using decision trees [20] and Bayesian networks [2, 28]. Spatial data could be modeled with a spatial decomposition tree [8, 27]. A set of non-linearly correlated continuous variables could be modeled using *copulas* [19, 23]. These models are restricted by the type of distributions and by computational issues, severely limiting the synthetic data's fidelity.

The development of generative models using VAEs and, subsequently, GANs and their numerous extensions [1, 11, 29, 26], has been very appealing due to the performance and flexibility offered in representing data. GANs are also used in generating tabular data, especially healthcare records; for example, [25] uses GANs to generate continuous time-series medical records and [4] proposes the generation of discrete tabular data using GANs. `medGAN` [6] combines an auto-encoder and a GAN to generate heterogeneous non-time-series continuous and/or binary data. `ehrGAN` [5] generates augmented medical records. `tableGAN` [18] tries to solve the problem of generating synthetic data using a convolutional neural network which optimizes the label column's quality; thus, generated data can be used to train classifiers. `PATE-GAN` [14] generates differentially private synthetic data.

## 3 Challenges with GANs in Tabular Data Generation Task

The task of synthetic data generation task requires training a data synthesizer $G$ learnt from a table $\mathbf{T}$ and then using $G$ to generate a synthetic table $\mathbf{T}_{syn}$. A table $\mathbf{T}$ contains $N_c$ continuous columns $\{C_1, \ldots, C_{N_c}\}$ and $N_d$ discrete columns $\{D_1, \ldots, D_{N_d}\}$, where each column is considered to be a random variable. These random variables follow an unknown joint distribution $\mathbb{P}(C_{1:N_c}, D_{1:N_d})$. One row $\mathbf{r}_j = \{c_{1,j}, \ldots, c_{N_c,j}, d_{1,j}, \ldots, d_{N_d,j}\}$, $j \in \{1, \ldots, n\}$, is one observation from the joint distribution. $\mathbf{T}$ is partitioned into training set $\mathbf{T}_{train}$ and test set $\mathbf{T}_{test}$. After training $G$ on $\mathbf{T}_{train}$, $\mathbf{T}_{syn}$ is constructed by independently sampling rows using $G$. We evaluate the efficacy of a generator along 2 axes. (1) *Likelihood fitness*: Do columns in $\mathbf{T}_{syn}$ follow the same joint distribution as $\mathbf{T}_{train}$? (2) *Machine learning efficacy*: When training a classifier or a regressor to predict one column using other columns as features, can such classifier or regressor learned from $\mathbf{T}_{syn}$ achieve a similar performance on $\mathbf{T}_{test}$, as a model learned on $\mathbf{T}_{train}$?

Several unique properties of tabular data challenge the design of a GAN model.

**Mixed data types.** Real-world tabular data consists of mixed types. To simultaneously generate a mix of discrete and continuous columns, GANs must apply both `softmax` and `tanh` on the output.

**Non-Gaussian distributions**: In images, pixels' values follow a Gaussian-like distribution, which can be normalized to $[-1, 1]$ using a min-max transformation. A `tanh` function is usually employed in the last layer of a network to output a value in this range. Continuous values in tabular data are usually non-Gaussian where min-max transformation will lead to vanishing gradient problem.

**Multimodal distributions.** We use kernel density estimation to estimate the number of modes in a column. We observe that $57/123$ continuous columns in our 8 real-world datasets have multiple modes. Srivastava et al. [21] showed that vanilla GAN couldn't model all modes on a simple 2D dataset; thus it would also struggle in modeling the multimodal distribution of continuous columns.

**Learning from sparse one-hot-encoded vectors.** When generating synthetic samples, a generative model is trained to generate a probability distribution over all categories using `softmax`, while the real data is represented in one-hot vector. This is problematic because a trivial discriminator can simply distinguish real and fake data by checking the distribution's sparseness instead of considering the overall realness of a row.

**Highly imbalanced categorical columns.** In our datasets we noticed that $636/1048$ of the categorical columns are highly imbalanced, in which the major category appears in more than $90\%$ of the rows. This creates severe mode collapse. Missing a minor category only causes tiny changes to the data distribution that is hard to be detected by the discriminator. Imbalanced data also leads to insufficient training opportunities for minor classes.

## 4 CTGAN Model

`CTGAN` is a GAN-based method to model tabular data distribution and sample rows from the distribution. In `CTGAN`, we invent the *mode-specific normalization* to overcome the non-Gaussian and multimodal distribution (Section 4.2). We design a *conditional generator* and *training-by-sampling* to deal with the imbalanced discrete columns (Section 4.3). And we use fully-connected networks and several recent techniques to train a high-quality model.

### 4.1 Notations

We define the following notations.

- $x_1 \oplus x_2 \oplus \ldots$: concatenate vectors $x_1, x_2, \ldots$
- $\texttt{gumbel}_\tau(x)$: apply Gumbel softmax[13] with parameter $\tau$ on a vector $x$
- $\texttt{leaky}_\gamma(x)$: apply a leaky ReLU activation on $x$ with leaky ratio $\gamma$
- $\texttt{FC}_{u \to v}(x)$: apply a linear transformation on a $u$-dim input to get a $v$-dim output.

We also use `tanh`, `ReLU`, `softmax`, `BN` for batch normalization [12], and `drop` for dropout [22].

### 4.2 Mode-specific Normalization

Properly representing the data is critical in training neural networks. Discrete values can naturally be represented as one-hot vectors, while representing continuous values with arbitrary distribution is non-trivial. Previous models [6, 18] use min-max normalization to normalize continuous values to $[-1, 1]$. In `CTGAN`, we design a *mode-specific* normalization to deal with columns with complicated distributions.

Figure 1 shows our mode-specific normalization for a continuous column. In our method, each column is processed independently. Each value is represented as a one-hot vector indicating the mode, and a scalar indicating the value within the mode. Our method contains three steps.

1. For each continuous column $C_i$, use variational Gaussian mixture model (VGM) [3] to estimate the number of modes $m_i$ and fit a Gaussian mixture. For instance, in Figure 1, the VGM finds three modes ($m_i = 3$), namely $\eta_1$, $\eta_2$ and $\eta_3$. The learned Gaussian mixture is $\mathbb{P}_{C_i}(c_{i,j}) = \sum_{k=1}^3 \mu_k \mathcal{N}(c_{i,j}; \eta_k, \phi_k)$ where $\mu_k$ and $\phi_k$ are the weight and standard deviation of a mode respectively.

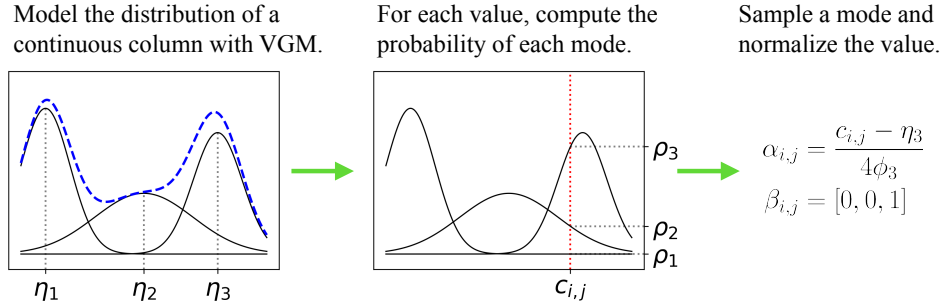

Figure 1: An example of mode-specific normalization.

2. For each value $c_{i,j}$ in $C_i$, compute the probability of $c_{i,j}$ coming from each mode. For instance, in Figure 1, the probability densities are $\rho_1, \rho_2, \rho_3$. The probability densities are computed as $\rho_k = \mu_k \mathcal{N}(c_{i,j}; \eta_k, \phi_k)$.

3. Sample one mode from given the probability density, and use the sampled mode to normalize the value. For example, in Figure 1, we pick the third mode given $\rho_1$, $\rho_2$ and $\rho_3$. Then we represent $c_{i,j}$ as a one-hot vector $\beta_{i,j} = [0, 0, 1]$ indicating the third mode, and a scalar $\alpha_{i,j} = \frac{c_{i,j} - \eta_3}{4\phi_3}$ to represent the value within the mode.

The representation of a row become the concatenation of continuous and discrete columns

$$\mathbf{r}_j = \alpha_{1,j} \oplus \beta_{1,j} \oplus \ldots \oplus \alpha_{N_c,j} \oplus \beta_{N_c,j} \oplus \mathbf{d}_{1,j} \oplus \ldots \oplus \mathbf{d}_{N_d,j},$$

where $\mathbf{d}_{i,j}$ is one-hot representation of a discrete value.

## 4.3 Conditional Generator and Training-by-Sampling

Traditionally, the generator in a GAN is fed with a vector sampled from a standard multivariate normal distribution (MVN). By training together with a *Discriminator* or *Critic* neural networks, one eventually obtains a deterministic transformation that maps the standard MVN into the distribution of the data. This method of training a generator does not account for the imbalance in the categorical columns. If the training data are randomly sampled during training, the rows that fall into the minor category will not be sufficiently represented, thus the generator may not be trained correctly. If the training data are resampled, the generator learns the resampled distribution which is different from the real data distribution. This problem is reminiscent of the "*class imbalance*" problem in discriminatory modeling - the challenge however is exacerbated since there is not a single column to balance and the real data distribution should be kept intact.

Specifically, the goal is to resample efficiently in a way that all the categories from discrete attributes are sampled evenly (but not necessary uniformly) during the training process, and to recover the (not-resampled) real data distribution during test. Let $k^*$ be the value from the $i^*$th discrete column $D_{i^*}$ that has to be matched by the generated samples $\hat{\mathbf{r}}$, then the generator can be interpreted as the conditional distribution of rows given that particular value at that particular column, i.e. $\hat{\mathbf{r}} \sim \mathbb{P}_{\mathcal{G}}(\text{row}|D_{i*} = k^*)$. For this reason, in this paper we name it *Conditional generator*, and a GAN built upon it is referred to as *Conditional GAN*.

Integrating a conditional generator into the architecture of a GAN requires to deal with the following issues: 1) it is necessary to devise a representation for the condition as well as to prepare an input for it, 2) it is necessary for the generated rows to preserve the condition as it is given, and 3) it is necessary for the conditional generator to learn the real data conditional distribution, i.e. $\mathbb{P}_{\mathcal{G}}(\text{row}|D_{i*} = k^*) = \mathbb{P}(\text{row}|D_{i*} = k^*)$, so that we can reconstruct the original distribution as

$$\mathbb{P}(\text{row}) = \sum_{k \in D_{i*}} \mathbb{P}_{\mathcal{G}}(\text{row}|D_{i*} = k^*)\mathbb{P}(D_{i*} = k).$$

We present a solution that consists of three key elements, namely: the *conditional vector*, the generator loss, and the *training-by-sampling* method.

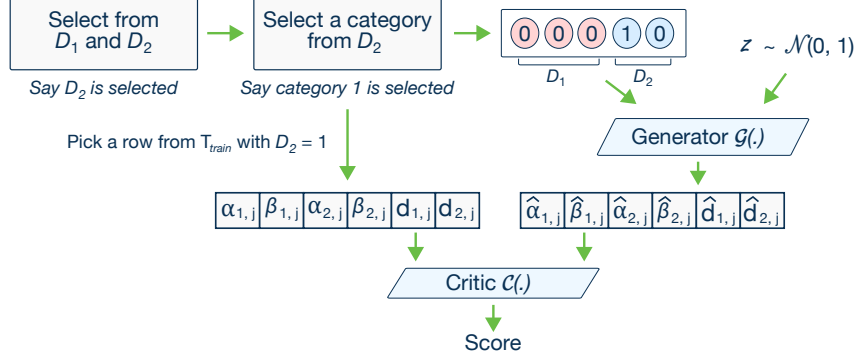

Figure 2: `CTGAN` model. The conditional generator can generate synthetic rows conditioned on one of the discrete columns. With training-by-sampling, the *cond* and training data are sampled according to the log-frequency of each category, thus `CTGAN` can evenly explore all possible discrete values.

**Conditional vector.** We introduce the vector *cond* as the way for indicating the condition $(D_{i^*} = k^*)$. Recall that all the discrete columns $D_1, \ldots, D_{N_d}$ end up as one-hot vectors $\mathbf{d}_1, \ldots, \mathbf{d}_{N_d}$ such that the $i$th one-hot vector is $\mathbf{d}_i = [\mathbf{d}_i^{(k)}]$, for $k = 1, \ldots, |D_i|$. Let $\mathbf{m}_i = [\mathbf{m}_i^{(k)}]$, for $k = 1, \ldots, |D_i|$ be the $i$th *mask* vector associated to the $i$th one-hot vector $\mathbf{d}_i$. Hence, the condition can be expressed in terms of these mask vectors as

$$\mathbf{m}_i^{(k)} = \begin{cases} 1 & \text{if } i = i^* \text{ and } k = k^*, \\ 0 & \text{otherwise.} \end{cases}$$

Then, define the vector *cond* as $cond = \mathbf{m}_1 \oplus \ldots \oplus \mathbf{m}_{N_d}$. For instance, for two discrete columns, $D_1 = \{1, 2, 3\}$ and $D_2 = \{1, 2\}$, the condition $(D_2 = 1)$ is expressed by the mask vectors $\mathbf{m}_1 = [0, 0, 0]$ and $\mathbf{m}_2 = [1, 0]$; so $cond = [0, 0, 0, 1, 0]$.

**Generator loss.** During training, the conditional generator is free to produce any set of one-hot discrete vectors $\{\hat{\mathbf{d}}_1, \ldots, \hat{\mathbf{d}}_{N_d}\}$. In particular, given the condition $(D_{i^*} = k^*)$ in the form of *cond* vector, nothing in the feed-forward pass prevents from producing either $\hat{\mathbf{d}}_{i^*}^{(k^*)} = 0$ or $\hat{\mathbf{d}}_{i^*}^{(k)} = 1$ for $k \neq k^*$. The mechanism proposed to enforce the conditional generator to produce $\hat{\mathbf{d}}_{i^*} = \mathbf{m}_{i^*}$ is to penalize its loss by adding the cross-entropy between $\mathbf{m}_{i^*}$ and $\hat{\mathbf{d}}_{i^*}$, averaged over all the instances of the batch. Thus, as the training advances, the generator learns to make an exact copy of the given $\mathbf{m}_{i^*}$ into $\hat{\mathbf{d}}_{i^*}$.

**Training-by-sampling.** The output produced by the conditional generator must be assessed by the critic, which estimates the distance between the learned conditional distribution $\mathbb{P}_{\mathcal{G}}(\text{row}|cond)$ and the conditional distribution on real data $\mathbb{P}(\text{row}|cond)$. The sampling of real training data and the construction of *cond* vector should comply to help critic estimate the distance. Properly sample the *cond* vector and training data can help the model evenly explore all possible values in discrete columns. For our purposes, we propose the following steps:

1. Create $N_d$ zero-filled mask vectors $\mathbf{m}_i = [\mathbf{m}_i^{(k)}]_{k=1\ldots|D_i|}$, for $i = 1, \ldots, N_d$, so the $i$th mask vector corresponds to the $i$th column, and each component is associated to the category of that column.
2. Randomly select a discrete column $D_i$ out of all the $N_d$ discrete columns, with equal probability. Let $i^*$ be the index of the column selected. For instance, in Figure 2, the selected column was $D_2$, so $i^* = 2$.
3. Construct a PMF across the range of values of the column selected in 2, $D_{i^*}$, such that the probability mass of each value is the logarithm of its frequency in that column.
4. Let $k^*$ be a randomly selected value according to the PMF above. For instance, in Figure 2, the range $D_2$ has two values and the first one was selected, so $k^* = 1$.
5. Set the $k^*$th component of the $i^*$th mask to one, i.e. $\mathbf{m}_{i^*}^{(k^*)} = 1$.
6. Calculate the vector $cond = \mathbf{m}_1 \oplus \cdots \mathbf{m}_{i^*} \oplus \mathbf{m}_{N_d}$. For instance, in Figure 2, we have the masks $\mathbf{m}_1 = [0, 0, 0]$ and $\mathbf{m}_{2^*} = [1, 0]$, so $cond = [0, 0, 0, 1, 0]$.

## 4.4 Network Structure

Since columns in a row do not have local structure, we use fully-connected networks in generator and critic to capture all possible correlations between columns. Specifically, we use two fully-connected hidden layers in both generator and critic. In generator, we use batch-normalization and Relu activation function. After two hidden layers, the synthetic row representation is generated using a mix activation functions. The scalar values $\alpha_i$ is generated by $tanh$, while the mode indicator $\beta_i$ and discrete values $\mathbf{d}_i$ is generated by gumbel softmax. In critic, we use leaky relu function and dropout on each hidden layer.

Finally, the conditional generator $\mathcal{G}(z, cond)$ can be formally described as

$$
\begin{cases}
h_0 = z \oplus cond \\
h_1 = h_0 \oplus \texttt{ReLU}(\texttt{BN}(\texttt{FC}_{|cond|+|z|\to256}(h_0))) \\
h_2 = h_1 \oplus \texttt{ReLU}(\texttt{BN}(\texttt{FC}_{|cond|+|z|+256\to256}(h_1))) \\
\hat{\alpha}_i = \texttt{tanh}(\texttt{FC}_{|cond|+|z|+512\to1}(h_2)) & 1 \le i \le N_c \\
\hat{\beta}_i = \texttt{gumbel}_{0.2}(\texttt{FC}_{|cond|+|z|+512\to m_i}(h_2)) & 1 \le i \le N_c \\
\hat{\mathbf{d}}_i = \texttt{gumbel}_{0.2}(\texttt{FC}_{|cond|+|z|+512\to|D_i|}(h_2)) & 1 \le i \le N_d
\end{cases}
$$

We use the PacGAN [17] framework with 10 samples in each pac to prevent mode collapse. The architecture of the critic (with pac size 10) $\mathcal{C}(\mathbf{r}_1, \ldots, \mathbf{r}_{10}, cond_1, \ldots, cond_{10})$ can be formally described as

$$
\begin{cases}
h_0 = \mathbf{r}_1 \oplus \ldots \oplus \mathbf{r}_{10} \oplus cond_1 \oplus \ldots \oplus cond_{10} \\
h_1 = \texttt{drop}(\texttt{leaky}_{0.2}(\texttt{FC}_{10|\mathbf{r}|+10|cond|\to256}(h_0))) \\
h_2 = \texttt{drop}(\texttt{leaky}_{0.2}(\texttt{FC}_{256\to256}(h_1))) \\
\mathcal{C}(\cdot) = \texttt{FC}_{256\to1}(h_2)
\end{cases}
$$

We train the model using WGAN loss with gradient penalty [11]. We use Adam optimizer with learning rate $2 \cdot 10^{-4}$.

## 4.5 TVAE Model

Variational autoencoder is another neural network generative model. We adapt VAE to tabular data by using the same preprocessing and modifying the loss function. We call this model `TVAE`. In `TVAE`, we use two neural networks to model $p_\theta(\mathbf{r}_j|z_j)$ and $q_\phi(z_j|\mathbf{r}_j)$, and train them using evidence lower-bound (ELBO) loss [15].

The design of the network $p_\theta(\mathbf{r}_j|z_j)$ that needs to be done differently so that the probability can be modeled accurately. In our design, the neural network outputs a joint distribution of $2N_c + N_d$ variables, corresponding to $2N_c + N_d$ variables $\mathbf{r}_j$. We assume $\alpha_{i,j}$ follows a Gaussian distribution with different means and variance. All $\beta_{i,j}$ and $\mathbf{d}_{i,j}$ follow a categorical PMF. Here is our design.

$$
\begin{cases}
h_1 = \texttt{ReLU}(\texttt{FC}_{128\to128}(z_j)) \\
h_2 = \texttt{ReLU}(\texttt{FC}_{128\to128}(h_1)) \\
\bar{\alpha}_{i,j} = \texttt{tanh}(\texttt{FC}_{128\to1}(h_2)) & 1 \le i \le N_c \\
\hat{\alpha}_{i,j} \sim \mathcal{N}(\bar{\alpha}_{i,j}, \delta_i) & 1 \le i \le N_c \\
\hat{\beta}_{i,j} \sim \texttt{softmax}(\texttt{FC}_{128\to m_i}(h_2)) & 1 \le i \le N_c \\
\hat{\mathbf{d}}_{i,j} \sim \texttt{softmax}(\texttt{FC}_{128\to|D_i|}(h_2)) & 1 \le i \le N_d \\
p_\theta(\mathbf{r}_j|z_j) = \prod_{i=1}^{N_c} \mathbb{P}(\hat{\alpha}_{i,j} = \alpha_{i,j}) \prod_{i=1}^{N_c} \mathbb{P}(\hat{\beta}_{i,j} = \beta_{i,j}) \prod_{i=1}^{N_d} \mathbb{P}(\hat{\alpha}_{i,j} = \alpha_{i,j})
\end{cases}
$$

Here $\hat{\alpha}_{i,j}, \hat{\beta}_{i,j}, \hat{\mathbf{d}}_{i,j}$ are random variables. And $p_\theta(\mathbf{r}_j|z_j)$ is the joint distribution of these variables. In $p_\theta(\mathbf{r}_j|z_j)$, weight matrices and $\delta_i$ are parameters in the network. These parameters are trained using gradient descent.

The modeling for $q_\phi(z_j|\mathbf{r}_j)$ is similar to conventional VAE.

$$
\begin{cases}
h_1 = \texttt{ReLU}(\texttt{FC}_{|\mathbf{r}_j|\to128}(\mathbf{r}_j)) \\
h_2 = \texttt{ReLU}(\texttt{FC}_{128\to128}(h_1)) \\
\mu = \texttt{FC}_{128\to128}(h_2) \\
\sigma = \exp(\frac{1}{2}\texttt{FC}_{128\to128}(h_2)) \\
q_\phi(z_j|\mathbf{r}_j) \sim \mathcal{N}(\mu, \sigma\mathbf{I})
\end{cases}
$$

`TVAE` is trained using Adam with learning rate 1e-3.

## 5 Benchmarking Synthetic Data Generation Algorithms

There are multiple deep learning methods for modeling tabular data. We noticed that all methods and their corresponding papers neither employed the same datasets nor were evaluated under similar metrics. This fact made comparison challenging and did not allow for identifying each method's weaknesses and strengths *vis-a-vis* the intrinsic challenges presented when modeling tabular data. To address this, we developed a comprehensive benchmarking suite.

### 5.1 Baselines and Datasets

In our benchmarking suite, we have baselines that consist of Bayesian networks (`CLBN` [7], `PrivBN` [28]), and implementations of current deep learning approaches for synthetic data generation (`MedGAN` [6], `VeeGAN` [21], `TableGAN` [18]). We compare `TVAE` and `CTGAN` with these baselines.

Our benchmark contains 7 simulated datasets and 8 real datasets.

**Simulated data:** We handcrafted a data oracle $\mathcal{S}$ to represent a known joint distribution, then sample $\mathbf{T}_{train}$ and $\mathbf{T}_{test}$ from $\mathcal{S}$. This oracle is either a Gaussian mixture model or a Bayesian network. We followed procedures found in [21] to generate `Grid` and `Ring` Gaussian mixture oracles. We added random offset to each mode in `Grid` and called it `GridR`. We picked 4 well known Bayesian networks - `alarm`, `child`, `asia`, `insurance`,[3] - and constructed Bayesian network oracles.

**Real datasets**: We picked 6 commonly used machine learning datasets from UCI machine learning repository [9], with features and label columns in a tabular form - `adult`, `census`, `covertype`, `intrusion` and `news`. We picked `credit` from Kaggle. We also binarized $28 \times 28$ the MNIST [16] dataset and converted each sample to 784 dimensional feature vector plus one label column to mimic high dimensional binary data, called `MNIST28`. We resized the images to $12 \times 12$ and used the same process to generate a dataset we call `MNIST12`. All in all there are 8 real datasets in our benchmarking suite.

### 5.2 Evaluation Metrics and Framework

Given that evaluation of generative models is not a straightforward process, where different metrics yield substantially diverse results [24], our benchmarking suite evaluates multiple metrics on multiple datasets. Simulated data come from a known probability distribution and for them we can evaluate the generated synthetic data via *likelihood fitness metric*. For real datasets, there is a machine learning task and we evaluate synthetic data generation method via *machine learning efficacy*. Figure 3 illustrates the evaluation framework.

**Likelihood fitness metric**: On simulated data, we take advantage of simulated data oracle $\mathcal{S}$ to compute the *likelihood fitness* metric. We compute the likelihood of $T_{syn}$ on $\mathcal{S}$ as $\mathcal{L}_{syn}$. $\mathcal{L}_{syn}$ prefers overfited models. To overcome this issue, we use another metric, $\mathcal{L}_{test}$. We retrain the simulated data oracle $\mathcal{S}'$ using $\mathbf{T}_{syn}$. $\mathcal{S}'$ has the same structure but different parameters than $\mathcal{S}$. If $\mathcal{S}$ is a Gaussian mixture model, we use the same number of Gaussian components and retrain the mean and covariance of each component. If $\mathcal{S}$ is a Bayesian network, we keep the same graphical structure and learn a new conditional distribution on each edge. Then $\mathcal{L}_{test}$ is the likelihood of $\mathbf{T}_{test}$ on $\mathcal{S}'$. This metric overcomes the issue in $\mathcal{L}_{syn}$. It can detect mode collapse. But this metric introduces the prior knowledge of the structure of $\mathcal{S}'$ which is not necessarily encoded in $\mathbf{T}_{syn}$.

**Machine learning efficacy**: For a real dataset, we cannot compute the likelihood fitness, instead we evaluate the performance of using synthetic data as training data for machine learning. We train prediction models on $\mathbf{T}_{syn}$ and test prediction models using $\mathbf{T}_{test}$. We evaluate the performance of classification tasks using accuracy and F1, and evaluate the regression tasks using $R^2$. For each dataset, we select classifiers or regressors that achieve reasonable performance on each data. (Models and hyperparameters can be found in supplementary material as well as our benchmark framework.) Since we are not trying to pick the best classification or regression model, we take the the average performance of multiple prediction models to evaluate our metric for $G$.

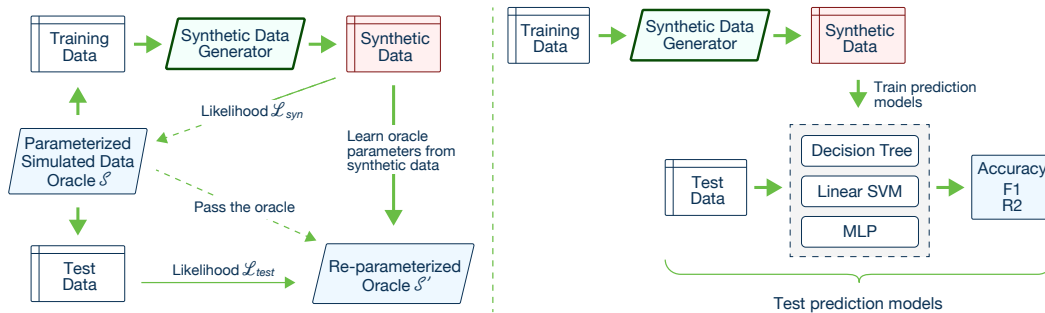

Figure 3: Evaluation framework on simulated data (left) and real data (right).

Table 2: Benchmark results over three sets of experiments, namely Gaussian mixture simulated data (GM Sim.), Bayesian network simulated data (BN Sim.), and real data. For GM Sim. and BN Sim., we report the average of each metric. For real datasets, we report average F1 for classification tasks and $\bar{R}^2$ for regression tasks respectively.

| | GM Sim. | | BN Sim. | | Real | |
| Method | $\mathcal{L}_{syn}$ | $\mathcal{L}_{test}$ | $\mathcal{L}_{syn}$ | $\mathcal{L}_{test}$ | clf | reg |
|---|---|---|---|---|---|---|
| Identity | -2.61 | -2.61 | -9.33 | -9.36 | 0.743 | 0.14 |
| CLBN | -3.06 | -7.31 | -10.66 | -9.92 | 0.382 | -6.28 |
| PrivBN | -3.38 | -12.42 | -12.97 | -10.90 | 0.225 | -4.49 |
| MedGAN | -7.27 | -60.03 | -11.14 | -12.15 | 0.137 | -8.80 |
| VEEGAN | -10.06 | -4.22 | -15.40 | -13.86 | 0.143 | -6.5e6 |
| TableGAN | -8.24 | -4.12 | -11.84 | -10.47 | 0.162 | -3.09 |
| TVAE | **-2.65** | -5.42 | **-6.76** | **-9.59** | **0.519** | **-0.20** |
| CTGAN | -5.72 | **-3.40** | -11.67 | -10.60 | 0.469 | -0.43 |

## 5.3 Benchmarking Results

We evaluated CLBN, PrivBN, MedGAN, VeeGAN, TableGAN, CTGAN, and TVAE using our benchmark framework. We trained each model with a batch size of $500$. Each model is trained for $300$ epochs. Each epoch contains $N/batch\_size$ steps where $N$ is the number of rows in the training set. We posit that for any dataset, across any metrics except $\mathcal{L}_{syn}$, the best performance is achieved by $\mathbf{T}_{train}$. Thus we present the Identity method which outputs $\mathbf{T}_{train}$.

We summarize the benchmark results in Table 2. Full results table can be found in Supplementary Material. For simulated data from Gaussian mixture, CLBN and PrivBN suffer because continuous numeric data has to be discretized before modeling using Bayesian networks. MedGAN, VeeGAN, and TableGAN all suffer from mode collapse. With mode-specific normalization, our model performs well on these 2-dimensional continuous datasets.

On simulated data from Bayesian networks, CLBN and PrivBN have a natural advantage. Our CTGAN achieves slightly better performance than MedGAN and TableGAN. Surprisingly, TableGAN works well on these datasets, despite considering discrete columns as continuous values. One possible reasoning for this is that in our simulated data, most variables have fewer than 4 categories, so conversion does not cause serious problems.

On real datasets, TVAE and CTGAN outperform CLBN and PrivBN, whereas other GAN models cannot get as good a result as Bayesian networks. With respect to large scale real datasets, learning a high-quality Bayesian network is difficult. So models trained on CLBN and PrivBN synthetic data are $36.1\%$ and $51.8\%$ worse than models trained on real data.

TVAE outperforms CTGAN in several cases, but GANs do have several favorable attributes, and this does not indicate that we should always use VAEs rather than GANs to model tables. The generator in GANs does not have access to real data during the entire training process; thus, we can make CTGAN achieve differential privacy [14] easier than TVAE.

### 5.4 Ablation Study

We did an ablation study to understand the usefulness of each of the components in our model. Table 3 shows the results from the ablation study.

*Mode-specific normalization.* In CTGAN, we use variational Gaussian mixture model (VGM) to normalize continuous columns. We compare it with (1) GMM5: Gaussian mixture model with 5 modes, (2) GMM10: Gaussian mixture model with 10 modes, and (3) MinMax: min-max normalization to $[-1, 1]$. Using GMM slightly decreases the performance while min-max normalization gives the worst performance.

*Conditional generator and training-by-sampling*: We successively remove these two components. (1) w/o S.: we first disable training-by-sampling in training, but the generator still gets a condition vector and its loss function still has the cross-entropy term. The condition vector is sampled from training data frequency instead of log frequency. (2) w/o C.: We further remove the condition vector in the generator. These ablation results show that both training-by-sampling and conditional generator are critical for imbalanced datasets. Especially on highly imbalanced dataset such as credit, removing training-by-sampling results in $0\%$ on F1 metric.

*Network architecture:* In the paper, we use WGANGP+PacGAN. Here we compare it with three alternatives, WGANGP only, vanilla GAN loss only, and vanilla GAN + PacGAN. We observe that WGANGP is more suitable for synthetic data task than vanilla GAN, while PacGAN is helpful for vanilla GAN loss but not as important for WGANGP.

Table 3: Ablation study results on mode-specific normalization, conditional generator and training-by-sampling module, as well as the network architecture. The absolute performance change on real classification datasets (excluding MNIST) is reported.

| | Mode-specific Normalization | | | Generater | | Network Architechture | | |
|---|---|---|---|---|---|---|---|---|
| **Model** | GMM5 | GMM10 | MinMax | w/o S. | w/o C. | GAN | WGANGP | GAN+PacGAN |
| **Performance** | -4.1% | -8.6% | -25.7% | -17.8% | -36.5% | -6.5% | +1.75% | -5.2% |

## 6 Conclusion

In this paper we attempt to find a flexible and robust model to learn the distribution of columns with complicated distributions. We observe that none of the existing deep generative models can outperform Bayesian networks which discretize continuous values and learn greedily. We show several properties that make this task unique and propose our CTGAN model. Empirically, we show that our model can learn a better distributions than Bayesian networks. Mode-specific normalization can convert continuous values of arbitrary range and distribution into a bounded vector representation suitable for neural networks. And our conditional generator and training-by-sampling can over come the imbalance training data issue. Furthermore, we argue that the conditional generator can help generate data with a specific discrete value, which can be used for data augmentation. As future work, we would derive a theoretical justification on why GANs can work on a distribution with both discrete and continuous data.

## Acknowledgements

This paper is partially supported by the National Science Foundation Grants ACI-1443068. We (authors from MIT) also acknowledge generous support provided by Accenture for the synthetic data generation project. Dr. Cuesta-Infante is funded by the Spanish Government research fundings RTI2018-098743-B-I00 (MICINN/FEDER) and Y2018/EMT-5062 (Comunidad de Madrid).

## Footnotes

[1] Our `CTGAN` model is open-sourced at `https://github.com/DAI-Lab/CTGAN`

[2] Our benchmark can be found at `https://github.com/DAI-Lab/SDGym`.

[3]The structure of Bayesian networks can be found at `http://www.bnlearn.com/bnrepository/`.

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
