[Supplementary Material · sup.pdf]

# Supplementary Material: Modeling Tabular data using Conditional GAN

**Lei Xu**  **Maria Skoularidou**  **Alfredo Cuesta-Infante**  **Kalyan Veeramachaneni**

## 1   Dataset Details

The statistical information of simulated and real data is in Table 1. The raw data of 8 real datasets are avialable online.

- Adult: `http://archive.ics.uci.edu/ml/datasets/adult`
- Census: `https://archive.ics.uci.edu/ml/datasets/census+income`
- Covertype: `https://archive.ics.uci.edu/ml/datasets/covertype`
- Credit: `https://www.kaggle.com/mlg-ulb/creditcardfraud`
- Intrusion: `http://archive.ics.uci.edu/ml/datasets/kdd+cup+1999+data`
- MNIST: `http://yann.lecun.com/exdb/mnist/index.html`
- News: `https://archive.ics.uci.edu/ml/datasets/online+news+popularity`

For each dataset, we select a few classifiers or regressors which give reasonable performance on such dataset shown in Table 2.

Table 1: Datasets in our benchmark.

| Simulated Data | | | | | Real Data | | | | | |
|---|---|---|---|---|---|---|---|---|---|---|
| name | #train/test | #C | #B | #M | name | #train/test | #C | #B | #M | task |
| grid | 10k/10k | 2 | 0 | 0 | adult | 23k/10k | 6 | 2 | 7 | C |
| gridr | 10k/10k | 2 | 0 | 0 | census | 200k/100k | 7 | 3 | 31 | C |
| ring | 10k/10k | 2 | 0 | 0 | covertype | 481k/100k | 10 | 44 | 1 | C |
| asia | 10k/10k | 0 | 8 | 0 | credit | 264k/20k | 29 | 1 | 0 | C |
| alarm | 10k/10k | 0 | 13 | 24 | intrusion | 394k/100k | 26 | 5 | 10 | C |
| child | 10k/10k | 0 | 8 | 12 | mnist12 | 60k/10k | 0 | 144 | 1 | C |
| insurance | 10k/10k | 0 | 8 | 19 | mnist28 | 60k/10k | 0 | 784 | 1 | C |
| | | | | | news | 31k/8k | 45 | 14 | 0 | R |

#C, #B, and #M mean number of continuous columns, binary columns and multi-class discrete columns respectively. C and R in task mean classification and regression respectively.

Table 2: Classifiers and regressors selected for each real dataset and corresponding performance.

| dataset | name | accuracy | f1 | macro_f1 | micro_f1 | r2 |
|---|---|---|---|---|---|---|
| adult | Adaboost (estimator=50) | 86.07% | 68.03% | | | |
| | Decision Tree (depth=20) | 79.84% | 65.77% | | | |
| | Logistic Regression | 79.53% | 66.06% | | | |
| | MLP (50) | 85.06% | 67.57% | | | |
| census | Adaboost (estimator=50) | 95.22% | 50.75% | | | |
| | Decision Tree (depth=30) | 90.57% | 44.97% | | | |
| | MLP (100) | 94.30% | 52.43% | | | |
| covtype | Decision Tree (depth=30) | 82.25% | | 73.62% | 82.25% | |
| | MLP (100) | 70.06% | | 56.78% | 70.06% | |
| credit | Adaboost (estimator=50) | 99.93% | 76.00% | | | |
| | Decision Tree (depth=30) | 99.89% | 66.67% | | | |
| | MLP (100) | 99.92% | 73.31% | | | |
| intrusion | Decision Tree (depth=30) | 99.91% | | 85.82% | 99.91% | |
| | MLP (100) | 99.93% | | 86.65% | 99.93% | |
| mnist12 | Decision Tree (depth=30) | 84.10% | | 83.88% | 84.10% | |
| | Logistic Regression | 87.29% | | 87.11% | 87.29% | |
| | MLP (100) | 94.40% | | 94.34% | 94.40% | |
| mnist28 | Decision Tree (depth=30) | 86.08% | | 85.89% | 86.08% | |
| | Logistic Regression | 91.42% | | 91.29% | 91.42% | |
| | MLP (100) | 97.28% | | 97.26% | 97.28% | |
| news | Linear Regression | | | | | 0.1390 |
| | MLP (100) | | | | | 0.1492 |

Table 3: Benchmark results over three sets of experiments, namely Gaussian mixture simulated data, Bayesian network simulated data, and real data. The number in the bracket is the rank of a method (lower better). It is computed as follows: For each set of experiment, (1) rank algorithms over all metrics in each set. (2) Take the average of all ranks of each algorithm. Get one score in range $[1, 7]$ for each algorithm. (3) Rank the score again.

| method | grid $\mathcal{L}_{syn}$ | grid $\mathcal{L}_{test}$ | gridr $\mathcal{L}_{syn}$ | gridr $\mathcal{L}_{test}$ | ring $\mathcal{L}_{syn}$ | ring $\mathcal{L}_{test}$ |
|---|---|---|---|---|---|---|
| Identity | -3.06 | -3.06 | -3.06 | -3.07 | -1.70 | -1.70 |
| CLBN(2) | -3.68 | -8.62 | -3.76 | -11.60 | -1.75 | **-1.70** |
| PrivBN(4) | -4.33 | -21.67 | -3.98 | -13.88 | -1.82 | -1.71 |
| MedGAN(7) | -10.04 | -62.93 | -9.45 | -72.00 | -2.32 | -45.16 |
| VEEGAN(6) | -9.81 | -4.79 | -12.51 | -4.94 | -7.85 | -2.92 |
| TableGAN(5) | -8.70 | -4.99 | -9.64 | -4.70 | -6.38 | -2.66 |
| TVAE(1) | **-2.86** | -11.26 | **-3.41** | **-3.20** | **-1.68** | -1.79 |
| TVAE(3) | -5.63 | **-3.69** | -8.11 | -4.31 | -3.43 | -2.19 |

| method | asia $\mathcal{L}_{syn}$ | asia $\mathcal{L}_{test}$ | alarm $\mathcal{L}_{syn}$ | alarm $\mathcal{L}_{test}$ | child $\mathcal{L}_{syn}$ | child $\mathcal{L}_{test}$ | insurance $\mathcal{L}_{syn}$ | insurance $\mathcal{L}_{test}$ |
|---|---|---|---|---|---|---|---|---|
| Identity | -2.23 | -2.24 | -10.3 | -10.3 | -12.0 | -12.0 | -12.8 | -12.9 |
| CLBN(3) | -2.44 | -2.27 | -12.4 | -11.2 | -12.6 | -12.3 | -15.2 | -13.9 |
| PrivBN(1) | **-2.28** | **-2.24** | -11.9 | -10.9 | -12.3 | **-12.2** | -14.7 | **-13.6** |
| MedGAN(5) | -2.81 | -2.59 | **-10.9** | -14.2 | -14.2 | -15.4 | -16.4 | -16.4 |
| VEEGAN(7) | -8.11 | -4.63 | -17.7 | -14.9 | -17.6 | -17.8 | -18.2 | -18.1 |
| TableGAN(6) | -3.64 | -2.77 | -12.7 | -11.5 | -15.0 | -13.3 | -16.0 | -14.3 |
| TVAE(2) | -2.31 | -2.27 | -11.2 | **-10.7** | **-12.3** | -12.3 | **-14.7** | -14.2 |
| TGAN(4) | -2.56 | -2.31 | -14.2 | -12.6 | -13.4 | -12.7 | -16.5 | -14.8 |

| method | adult F1 | census F1 | credit F1 | cover. Macro | intru. Macro | mnist12 Acc | mnist28 Acc | news $R^2$ |
|---|---|---|---|---|---|---|---|---|
| Identity | 0.669 | 0.494 | 0.720 | 0.652 | 0.862 | 0.886 | 0.916 | 0.14 |
| CLBN(3) | 0.334 | 0.310 | 0.409 | 0.319 | 0.384 | 0.741 | 0.176 | -6.28 |
| PrivBN(4) | 0.414 | 0.121 | 0.185 | 0.270 | 0.384 | 0.117 | 0.081 | -4.49 |
| MedGAN(6) | 0.375 | 0.000 | 0.000 | 0.093 | 0.299 | 0.091 | 0.104 | -8.80 |
| VEEGAN(6) | 0.235 | 0.094 | 0.000 | 0.082 | 0.261 | 0.194 | 0.136 | -6.5e6 |
| TableGAN(5) | 0.492 | 0.358 | 0.182 | 0.000 | 0.000 | 0.100 | 0.000 | -3.09 |
| TVAE(1) | **0.626** | 0.377 | 0.098 | **0.433** | 0.511 | **0.793** | **0.794** | **-0.20** |
| TGAN(1) | 0.601 | **0.391** | **0.672** | 0.324 | **0.528** | 0.394 | 0.371 | -0.43 |

**Algorithm 1:** Train `CTGAN` on step.

---

**Input:** Training data $\mathbf{T}_{train}$, Conditional generator and Critic parameters $\Phi_G$ and $\Phi_C$ respectively, batch size $m$, pac size $pac$.

**Result:** Conditional generator and Critic parameters $\Phi_G$, $\Phi_C$ updated.

1   Create masks $\{\mathbf{m}_1, \ldots, \mathbf{m}_{i^*}, \ldots, \mathbf{m}_{N_d}\}_j$,   for $1 \le j \le m$

2   Create condition vectors $cond_j$,   for $1 \le j \le m$ from masks      ▷ Create $m$ conditional vectors

3   Sample $\{z_j\} \sim \texttt{MVN}(0, \mathbf{I})$,   for $1 \le j \le m$

4   $\hat{\mathbf{r}}_j \leftarrow \texttt{Generator}(z_j, cond_j)$,   for $1 \le j \le m$           ▷ Generate fake data

5   Sample $\mathbf{r}_j \sim \texttt{Uniform}(\mathbf{T}_{train}|cond_j)$,   for $1 \le j \le m$         ▷ Get real data

6   $cond_k^{(pac)} \leftarrow cond_{k \times pac+1} \oplus \ldots \oplus cond_{k \times pac+pac}$,   for $1 \le k \le m/pac$ ▷ Conditional vector pacs

7   $\hat{\mathbf{r}}_k^{(pac)} \leftarrow \hat{\mathbf{r}}_{k \times pac+1} \oplus \ldots \oplus \hat{\mathbf{r}}_{k \times pac+pac}$,   for $1 \le k \le m/pac$         ▷ Fake data pacs

8   $\mathbf{r}_k^{(pac)} \leftarrow \mathbf{r}_{k \times pac+1} \oplus \ldots \oplus \mathbf{r}_{k \times pac+pac}$,   for $1 \le k \le m/pac$          ▷ Real data pacs

9   $\mathcal{L}_C \leftarrow \frac{1}{m/pac} \sum_{k=1}^{m/pac} \texttt{Critic}(\hat{\mathbf{r}}_k^{(pac)}, cond_k^{(pac)}) - \frac{1}{m/pac} \sum_{k=1}^{m/pac} \texttt{Critic}(\mathbf{r}_k^{(pac)}, cond_k^{(pac)})$

10   Sample $\rho_1, \ldots, \rho_{m/pac} \sim \texttt{Uniform}(0, 1)$

11   $\tilde{\mathbf{r}}_k^{(pac)} \leftarrow \rho_k \hat{\mathbf{r}}_k^{(pac)} + (1 - \rho_k)\mathbf{r}_k^{(pac)}$,   for $1 \le k \le m/pac$

12   $\mathcal{L}_{GP} \leftarrow \frac{1}{m/pac} \sum_{k=1}^{m/pac} (||\nabla_{\tilde{\mathbf{r}}_k^{(pac)}} \texttt{Critic}(\tilde{\mathbf{r}}_k^{(pac)}, cond_k^{(pac)})||_2 - 1)^2$         ▷ Gradient Penalty

13   $\Phi_C \leftarrow \Phi_C - 0.0002 \times \texttt{Adam}(\nabla_{\Phi_C}(\mathcal{L}_C + 10\mathcal{L}_{GP}))$

14   Regenerate $\hat{\mathbf{r}}_j$ following lines 1 to 7

15   $\mathcal{L}_G \leftarrow -\frac{1}{m/pac} \sum_{k=1}^{m/pac} \texttt{Critic}(\hat{\mathbf{r}}_k^{(pac)}, cond_k^{(pac)}) + \frac{1}{m} \sum_{j=1}^{m} \texttt{CrossEntropy}(\hat{\mathbf{d}}_{i^*,j}, \mathbf{m}_{i^*})$

16   $\Phi_G \leftarrow \Phi_G - 0.0002 \times \texttt{Adam}(\nabla_{\Phi_G}\mathcal{L}_G)$

---