[Reviews · NeurIPS 2019]

Reviewer 1



Originality: The main originality of the paper is a data transformation process applied to tabular data so a GAN can learn from them. This is definitely higher novel and can be potentially useful in similar situations involving such distributions. Apart from this, however, I feel that the authors are overclaiming a bit regarding several challenge/contributions: -C2 (L86): The choice of activation function certainly depends on the data format, listing that as a "challenge" seems a bit too much to me, unless the authors can point out non-trivial adaptations they made to address the problem (and apologize if I missed that...) -C4 (L98): again, hardly something new -C5 (L105): mode collapse is certainly well studied in literature (speaking of which, the authors should add references on newer approaches such as BourGAN), using an off-the-shelf solution (PacGAN), again, does not seem to me as an important contribution. Rephrasing the section and focus on the important contributions (C3, and perhaps C1) will make the contributions of the paper more clear, in my opinion. Quality: The paper is of high quality and the description of techniques is sound. However, I have major problems with the evaluation methods. In the case of this paper, when a very specific problem is being solved and a lot of domain specific designs are being applied, it will be surprising to see if the resulting performance is not better than more general-purpose models. A "comprehensive" evaluation over many models over many possible datasets is not going to convince me further that the paper is actually making a significant contributions. Instead, I think the experiment design should focus on analyzing the effectiveness of all the claimed contributions of the paper. Here are my suggestions for possible experiments to run: -a summary of results (no need for a giant table of everything) showing that the proposed model indeed can achieve state of the art performance on the task -detailed ablations over all the major claimed contributions of the paper: -architecture wise, instead of comparing (a pacgan based architecture) with veegan in a different setting, compare the major variations in gans addressing mode collapses, keeping everything else constant, and show that pacgan is the best architecture to use. -More importantly, I would want to see a in depth analysis about the effect of the data processing method. Why discretizing everything in the described way is the best solution? What are the effects of, for example, not doing the mode-specific normalization, or representing every column as continuous values (as in TableGAN)? -How effective is the training-by-sampling procedure proposed? What are the effect on performances using either of the two rejected possibilities? -How effective is conditioning? -I am not convinced by the TVAE vs TGAN experiments, more detailed analysis on that would be nice to have -In addition, showing some qualitative examples illustrating key improvements of the design choices might be nice to have, though not necessary. A few points in addition to experiment design: -I think the "theoretical justification on why GANs can work" mentioned in section 7 is actually very important. I am personally not convinced that GAN is the most suitable solution to this class of problems. It would be nice to provide more analysis on that front, to show that the design choices are not purely ad hoc. -I am not that familiar with all the metrics used to evaluate GANs, but perhaps some justifications on why the two evaluation metrics are used will be useful? Clarity: -Although the exposition is readable as it is, I feel that overall too much unnecessary explanations are used. There is no need to explicitly define everything using equations, especially for neural network architectures, where it's mostly common sense among the potential readers. I would encourage the authors to use figures for stuff such as architectures, especially in supplementary materials. It might also be a good idea to leave the detailed architecture description to supplemental, and focus more on the data transformation process (the figures in the supplemental is really helpful, for example) -The wording of "conditional" GAN is potentially misleading as as far as I know, conditional GAN already refers to something that exists in literature. Either cite those for similarities or use a different name, will make the article less confusing at first read. Significance: I believe the paper proposes a useful system for the task it is trying to solve, but even for the described task, I think more motivation is needed. Are you trying to generate novel yet coherent data, or are you just trying to generate data that preserves statistical properties of data yet removes important privacy information? (Those are different since the constraints to respect in the latter is a subset of those in the former, and if the purpose is the former, then I am not sure the described way of generating data using a sampled conditional vector is still fair) More importantly, I am not that sure if it is significant enough to the rest of the community. It would be helpful if the authors can provide more justification of how the introduced technique can be potentially useful to a more general set of problems. Without such discussions, I feel that the problem might be too domain specific for a venue like NeurIPS. Summary: I recognize the value of the work, but I think this work can benefit from more comprehensive evaluations and more justifications on why the proposed ideas is useful to a more general audience. I am inclining to reject the paper as it is, but will be willing to change my judgments if the authors can address some of my major concerns. ------------------------------ Post-rebuttal comments: I thank the authors for addressing all my major concerns. As I've said, I initially gave a 5 because without properly-designed evaluations, it is hard to judge whether the proposed new ideas are effective or not. I think the ablation studies have shown that the ideas do work and the paper is worth accepting. In addition to including ablation I would still encourage the authors to motivate the paper a bit better in the final version, talking about importance of tabular data and highlighting how the conceptual contributions can also apply beyond tab data (I think this is definitely the case). To clarify: these are not necessary, just some of the points that I personally feel could make this more engaging. Thanks again for the hard work!

Reviewer 2



Originality: This paper focuses on generating tabular data comprised of both continuous variables and categorical (discrete) variables. With some methodological ideas such as mode-specific normalization, and using conditional GAN to counter feature-imbalance, the proposed approach outperforms previous methods. In addition, related works are well cited. Quality: It is rather difficult to assess the importance of the methodological contributions of this paper without any ablation study. It would have been informative if I could see how influential mode-specific normalization and train-by-sampling were. It is also a weakening factor that TVAE mostly outperforms TGAN, the main model of this paper, in most tasks. Clarity: The paper is clearly written and easy to understand. Sufficient details are provided to reproduce the results with some effort. However, it would make the paper stronger if the source code and the benchmark datasets were publicly available. Significance: There are demands for generating synthetic tabular data, and this work will clearly be a milestone in that effort. It is doubtful if the community will use TGAN, but it is likely that they will at least use TVAE. ---------------------------------------------------------------------------------------- After reading the author feedback: I appreciate the ablation study. The argument for supporting TGAN's strength over TVAE is not unreasonable, but there is no empirical evidence to support the claim. Overall, I'm keeping the score to 6.

Reviewer 3



Originality: The paper has several novel contributions on tabular data representation learning and generative modeling. Quality: The technical contribution of the approaches are at a high level, although there is room for improvement when the connection with implementations is considered. Clarity: The paper is overall well-written, the notations and explanations are easy to follow. Significance: Tabular data is vastly understudied despite its popularity in real-world AI. Many fundamental challenges of tabular data learning are throughly studied in this paper, showing excellent results on generative modeling. I believe the topic of the paper is very important. Further comments/questions: - Reversible data transformation would be very important for discriminative modeling as well. Were they considered for critic models, for evaluation purposes? - How extensively neural network hyperparameters are optimized? Is there room for improvement through better hyperparameter optimization? More discussions would be valuable. - I couldn't understand why VAE outperforms GANs significantly in some cases. Are there fundamental optimization issues?

[Author Response · NeurIPS 2019]

Thank you all very much for carefully reviewing our paper. Since the common feedback was to add ablation studies, in this document, we first give ablation study results. We will add these to the paper. Then we explain the advantage of TGAN against TVAE. And finally, we address individual questions and comments.

**Ablation Study**

Table 1: Ablation study results. Numbers are absolute performance change on real datasets.

| Experiment | EXP1 | | | EXP2 | | EXP3 | | |
|---|---|---|---|---|---|---|---|---|
| **Model** | GMM5 | GMM10 | MinMax | w/o R. | w/o C. | GAN | WGANGP | GAN+PacGAN |
| **Performance** | -4.1% | -8.6% | -25.7% | -17.8% | -36.5% | -6.5% | +1.75% | -5.2% |

*EXP1. Mode-specific normalization*: In TGAN, we use variational Gaussian mixture model (VGM) to normalize continuous columns. We compare it with (1) GMM5: Gaussian mixture model with 5 modes, (2) GMM10: Gaussian mixture model with 10 modes, and (3) MinMax: min-max normalization to $[-1, 1]$. Using GMM slightly decreases the performance while min-max normalization gives the worst performance.

*EXP2. Resampling and condition vector*: We successively remove these two components. (1) w/o R.: we first disable resampling in training, but the generator still gets a condition vector and its loss function still has the cross-entropy term. The condition vector is sampled from training data frequency instead of log frequency. (2) w/o C.: We further remove the condition vector in the generator. These ablation results show that both resampling and condition vector are important for imbalanced datasets. Especially on highly imbalanced dataset such as credit, removing resampling results in $0\%$ on F1 metric.

*EXP3. WGANGP and PacGAN:* In the paper, we use WGANGP+PacGAN. Here we compare it with three alternatives, WGANGP only, vanilla GAN loss only, and vanilla GAN + PacGAN. We observe that WGANGP is more suitable for synthetic data task than vanilla GAN, while PacGAN is helpful for vanilla GAN loss but not as important for WGANGP.

**Why we put TGAN as the main method in our paper.**

TGAN has a few advantages over TVAE, namely (1) since the generator in GAN is not directly optimized by mean square error, it's easier to make it differentially private using existing frameworks like DPGAN and PATE-GAN. Empirically, we compute the distance between synthetic data and nearest neighbor in training data. We observe TGAN gets 13% larger distance than TVAE, while achieving the same accuracy or F1 score on the real data. We will add this to the paper. (2) TGAN is more flexible in the sense that they are capable of capturing interactions amongst variables through their architecture, though TVAE is not intrinsically capable of doing so. To this end, in scenarios where strong complex underlying structures are involved, TGAN shall outperform TVAE.

**Individual Comments and Questions**

To Reviewer #1: (1) We agree with you and per your advice, we will remove unnecessary equations and use figures for NN architectures. We will move figure about data transformation process to the main paper. (2) Regarding GANs evaluation, given that the data employed here are not images, visual fidelity (which is the most common metric in image generation tasks) could not be applied. Moreover, Fréchet distance inception (Heusel et. al, 2017) could not be employed to all synthetic datasets given that it is applied on Gaussian distributed data. Hence, we were restricted to employ metrics that could be applied and are widely used in mixed data scenarios (Theis et. al, 2015). (3) Regarding TGAN convergence, theoretical guarantees for GANs to convergence to a Nash equilibrium are hard to derive in the case of continuous data, and harder for mixed data scenarios. Nevertheless, we have empirically checked that our algorithm does converge for a fixed set of hyperparameters regardless of random seeds. We recognize a recent work by Daskalakis et. al, 2017 and would extend it to provide theoretical results in the future.

To Reviewer #2: (1) We conducted a set of ablation studies as proposed in the review (results presented above). We will add these to paper. (2) All our code, real and benchmark datasets are publicly available. We developed our code as an open benchmarking framework. We had our anonymous repository hosted on anonymous.4open.science. The URL is at line 2 in the supplementary material. We are really sorry that the service was down recently and it's back online now.

To Reviewer #3: (1) We tried several hyperparameter sets for TVAE and TGAN. We will add this detail to the paper. In future there is scope for improvement with hyperparameter tuning. (2) In our revision, we will add figures to show generated synthetic data. We will add more references to recent advances in GAN. We have an additional page to add these.

[Meta-Review · NeurIPS 2019]

This paper was considered thoroughly executed and well evaluated. The idea of applying GAN to modeling tabular data was deemed mixed: on the one hand, it is a very important data type that has received relatively little attention in the NeurIPS community, but on the other hand many of the core ideas and developments were deemed to be potentially specific to tabular data and thus not highly generalizable/informative/engaging to the NeurIPS community at large. The rebuttal and discussion of the ablation study have favorably improved reviewers opinions.